# Cracking the Monoubiquitin Code of Genetic Diseases

**DOI:** 10.3390/ijms21093036

**Published:** 2020-04-25

**Authors:** Raj Nayan Sewduth, Maria Francesca Baietti, Anna A. Sablina

**Affiliations:** 1VIB-KU Leuven Center for Cancer Biology, VIB, Herestraat 49, 3000 Leuven, Belgium; rajnayan.sewduth@kuleuven.vib.be (R.N.S.); mariafrancesca.baietti@kuleuven.vib.be (M.F.B.); 2Department of Oncology, KU Leuven, Herestraat 49, 3000 Leuven, Belgium

**Keywords:** ubiquitin system, genetic diseases, ubiquitin ligase, deubiquitinases, monoubiquitin signaling, vesicular trafficking, protein complex formation

## Abstract

Ubiquitination is a versatile and dynamic post-translational modification in which single ubiquitin molecules or polyubiquitin chains are attached to target proteins, giving rise to mono- or poly-ubiquitination, respectively. The majority of research in the ubiquitin field focused on degradative polyubiquitination, whereas more recent studies uncovered the role of single ubiquitin modification in important physiological processes. Monoubiquitination can modulate the stability, subcellular localization, binding properties, and activity of the target proteins. Understanding the function of monoubiquitination in normal physiology and pathology has important therapeutic implications, as alterations in the monoubiquitin pathway are found in a broad range of genetic diseases. This review highlights a link between monoubiquitin signaling and the pathogenesis of genetic disorders.

## 1. Introduction

Ubiquitination is a reversible post-translational modification process during which the highly conserved 76-aminoacid protein ubiquitin is conjugated to target proteins. Ubiquitin can be conjugated to a protein substrate via distinct mechanisms. Monoubiquitination is the attachment of a single ubiquitin molecule to a single lysine residue on a substrate protein, whereas multi-monoubiquitination is the conjugation of a single ubiquitin molecule to multiple lysine residues. Polyubiquitination occurs when ubiquitin molecules are attached end-to-end to a lysine residue on a substrate protein to form a poly-ubiquitin chain. In this case, ubiquitin molecules are conjugated through one of the seven lysine residues present on the ubiquitin itself (K6, K11, K27, K29, K33, K48, and 63) or the N-terminal methionine (M1). While most of the studies have described the role of specific polyubiquitination, such as K48-linked polyubiquitination for proteasomal degradation [1,2] or K63-linked polyubiquitination for vesicular trafficking [3], emerging evidences implicate monoubiquitination and multi-monoubiquitination in controlling numerous aspects of protein function, such as degradation, subcellular localization, and protein–protein interaction. In this review, we focus on the role of monoubiquitin conjugation in normal physiology and genetic disease.

### 1.1. Monoubiquitination in Protein Function

Even though K48-linked polyubiquitination is the “canonical” signal for proteasomal degradation, a handful of proteins are degraded following monoubiquitination. Degradative monoubiquitination targets proteins smaller than 150 amino acids with low structural disorder, whereas polyubiquitination recognizes proteins with highly disorganized structures independently of their size. Monoubiquitination-dependent substrates of proteasomal degradation are enriched for genes associated with carbohydrate transport and oxidative stress response pathways [4]. Because carbohydrate transporters are plasma membrane proteins, this finding is consistent with previous studies linking down-regulation of membrane receptors to monoubiquitination-mediated endocytosis [5]. The stability of ribosomal and proteasomal subunits is also regulated by monoubiquitination [6]. Multi-monoubiquitination could also act as a degradation signal [7]. For instance, multi-monoubiquitination of Cyclin B1 mediated by the Anaphase-promoting complex (APC/C) is sufficient to promote Cyclin B1 proteasomal degradation and mitotic exit [7].

The attachment of a single ubiquitin can serve as a signal for specific subcellular localization. It is well established that monoubiquitination triggers endocytosis of receptor tyrosine kinases (RTKs) and the NOTCH receptor 1 (NOTCH) from the plasma membrane [8,9]. Monoubiquitination targets the TNF receptor associated factor 4 (TRAF4) to cell–cell junctions where it is required to promote cell migration [10]. Dissociation of lipidated proteins from the plasma membrane could also be induced by ubiquitin conjugation [11]. Specifically, ubiquitin conjugated to K170 of the small GTPase Harvey rat sarcoma viral oncogene homolog (HRAS) sequesters its farnesyl and palmitoyl groups, impairing association of HRAS with the membrane [12]. Monoubiquitination is also involved in the control of nuclear/cytoplasmic shuttling. Vascular Endothelial Growth Factor (VEGF)-induced monoubiquitination of the actin-binding protein filamin B (FLNB) was shown to inhibit the nuclear translocation of Histone Deacetylase 7 (HDAC7). The monoubiquitinated form of filamin B binds to the nuclear localization signal of HDAC7, thereby transiently preventing its re-entry into the nucleus and mitigating the transcriptional repressor activity of HDAC7 on the genes required for VEGF-mediated responses [13]. In contrast, ubiquitin conjugation induces the nuclear export of the neddylation regulator, Defective In Cullin Neddylation (DCNL1). It was suggested that ubiquitin conjugation might act as a nuclear export signal that promotes the interaction with nuclear export machinery [14]. Furthermore, the monoubiquitination of the cancer stem cells marker CD133 (prominin-1) promotes its secretion into extracellular vesicles by facilitating the interaction of CD133 with the vesicular sorting protein tumor susceptibility gene 101 (TSG101) [15].

Multiple studies highlight a key function of monoubiquitination in controlling protein–protein interactions, both in positive and negative ways. Ubiquitin conjugation facilitates the interaction either by creating additional interaction surface or by attracting proteins containing ubiquitin-binding domains. Ubiquitin-binding domains have been classified into nearly 25 subfamilies. These domains are typically independently folded modular domains of up to 150 amino acids showing remarkable structural heterogeneity, which utilize diverse surfaces to contact ubiquitin [16,17]. Ubiquitin-binding activities are hard to detect and predict, both structurally and bioinformatically; thus, many of such ubiquitin-binding proteins are yet to be identified. Especially taking into account that a subset of ubiquitin interactors has no obvious structurally defined ubiquitin-binding domains, such as the disordered Proteasome Complex Subunit SEM1/DSS1, in which ubiquitin-binding sites are characterized by acidic and hydrophobic residues [18]. The low-affinity interactions between ubiquitin-binding domains and ubiquitin are required for rapid, timely, and reversible cellular responses to a particular stimulus. To illustrate this, insulin-like growth factor 1(IGF-I)-induced monoubiquitination of the insulin receptor substrate (IRS-2) favors its interaction with the ubiquitin-binding protein EPSIN1, thereby enhancing IRS-2-mediated signaling and cell proliferation induced by IGF-I [19]. T-cell stimulation leads to monoubiquitination of the paracaspase MALT1 (mucosa-associated lymphoid tissue protein-1) that promotes the formation and stabilization of the catalytically active MALT1 dimer [20]. On top of this, auto-monoubiquitination of several E3 ligases, such as neural precursor cell-expressed developmentally down-regulated 4 (NEDD4), E3 ubiquitin-protein ligase Itchy (ITCH), and X-linked inhibitor of apoptosis (XIAP), is implicated in the recruitment of substrates and regulation of the ligase activity. For instance, the auto-monoubiquitination of NEDD4 promotes the recruitment of its substrate epidermal growth factor receptor substrate 15 (EPS15) [21].

Monoubiquitination can also block protein–protein interactions by either creating steric hindrance or inducing an autoinhibitory conformation for proteins containing ubiquitin-binding domains. As an example for the first mechanism, the attachment of ubiquitin within the effector-binding domain of the RAS-like GTPase, RAS-related protein B (RALB) sterically inhibits its binding to the downstream effector exocyst complex component EXO84 [22]. Another example is the inhibitory effect of monoubiquitination on the aggregation of amyloid proteins. Specifically, N-terminal monoubiquitination of presynaptic neuronal proteins, tau^K18^ and α-synuclein, changes their aggregation properties, resulting in structurally distinct aggregate structures that are cleared through proteasomal degradation instead of accumulating in cells [23]. On the other hand, the monoubiquitination of proteins that contain ubiquitin-binding domains could impose their autoinhibitory conformation, thus providing an intrinsic switch-off mechanism. This mechanism is used to regulate the degradative activity of the proteasome. Ubiquitin conjugation to the proteasome regulatory subunit RPN10 blocks the binding between its ubiquitin-interacting motif and interactors containing ubiquitin conjugates [24]. Monoubiquitination of the endocytic proteins, EPS15, hepatocyte growth factor-regulated tyrosine kinase substrate (HGS or HRS), ubiquitin-associated and SH3 domain-containing protein A and B (UBASH3A and UNASH3B, also called STS2 and STS1) leads to intramolecular interactions between ubiquitin and their ubiquitin-binding domains, thereby preventing their binding to ubiquitinated targets and inhibiting their trafficking [25].

By differentially modifying the protein–protein interaction network, monoubiquitination provides precise and timely transmission of biological information [26]. To illustrate this, ubiquitin conjugation of the RAS-like GTPase RALB modifies its affinity for downstream effectors, inhibiting the interaction with EXO84 and facilitating its binding to exocyst complex component 2 (EXOC2 or SEC5). Thus, monoubiquitination within the effector-binding domain provides the switch for the dual functions of RALB in autophagy and innate immune responses [22]. Another example is the monoubiquitination of t-SNARE syntaxin 5 (SYN5) in early mitosis that disrupts SNARE complex formation. Subsequently, ubiquitinated SYN5 recruits p97/VCP (valosin-containing protein) to the mitotic Golgi fragments and promotes post-mitotic Golgi reassembly; thus, ubiquitin conjugation regulates Golgi membrane dynamics during the cell cycle [27].

Importantly, the attachment of a single ubiquitin molecule to specific lysine residues can generate diverse substrate–ubiquitin structures, leading to different functional outcomes. This is well-described for the RAS-like small GTPases that undergo monoubiquitination at several lysines. While monoubiquitination of the GTPase Ras-related protein RAB5 at K165 affects its GTPase activity, attachment of ubiquitin at K140 alters the ability of RAB5 to bind and activate its downstream effectors. Ubiquitin conjugation to specific lysines also differentially affects the activity and subcellular localization of the Rat sarcoma (RAS) proto-oncogenes [28]. Specifically, the monoubiquitination of RAS at K117 promotes GTP loading, whereas monoubiquitination at K147 impairs GTP hydrolysis activity, both resulting in RAS activation. On the other hand, monoubiquitination at K170 impairs RAS association to the membrane and downstream signaling [12,29,30,31]. Altogether, these examples demonstrate a fundamental role of monoubiquitination in controlling protein function.

### 1.2. Enzymes Controlling Monoubiquitination

The ubiquitination reaction involves three enzymes: ubiquitin activating enzyme (E1), ubiquitin conjugating enzymes (E2), and ubiquitin ligase (E3) enzymes [32]. Although there are only 2 E1s and 30–50 E2s, approximately 600–700 E3 ubiquitin ligases are encoded in the human genome, conferring target specificity to the ubiquitination reaction. The process is reversible as ubiquitin hydrolases, also called deubiquitinating enzymes (DUBs), can hydrolyze the covalently bound ubiquitin peptides.

E2 conjugating enzymes attach ubiquitin molecules to lysine residues on protein substrate or ubiquitin molecules, prompting mono- or poly-ubiquitination. E2s share a highly conserved catalytic core domain called the ubiquitin conjugation domain (UBC). The UBC contains a specific catalytic cysteine that forms a thioester bond with ubiquitin. Most of the studies have been focused on ubiquitin-conjugating enzymes 2S and 3A (UBE2S, UBE3A) and ubiquitin-conjugating enzyme E2 variant 1 (UEV1A) which are responsible for pro-degradative K48-linked polyubiquitination and UBE2N, which is important for pro-trafficking K63-linked polyubiquitination. These E2-conjugating enzymes also have intrinsic ability to form free polyubiquitin chains. On the other hand, several E2 ubiquitin ligases, such as ubiquitin-conjugating enzymes 2A, 2K, 2T, and 2W (UBE2A, UBE2K, UBE2T, and UBE2W) are only able to generate monoubiquitin chains. When E2s are charged with ubiquitin molecules, some of them exhibit enhanced affinity for co-factors containing ubiquitin-binding domains that confer specificity of ubiquitin linkages to specific substrates [33].

The substrate specificity is given by the E3 ubiquitin ligases that control the transfer of ubiquitin to a lysine residue on the substrate. The E3 ligases identify their protein substrate within the cellular pool of proteins, by recognizing specific protein sequences or chemical motifs in the substrate. The E3 ligases can be classified into RING (really interesting new gene), HECT (homology to E6AP C-terminus), and RING-related types [34]. The activity of the RING E3s is specified by a RING domain, which promotes ubiquitin transfer from the E2 to the substrate, whereas the substrate-recruiting module is responsible for the substrate recognition. The RING domain and the substrate-recruiting module of the RING E3 ligase can be found in a single polypeptide, as in the case of CBL (Casitas B-lineage lymphoma), or in separate subunits of a multi-complex E3s, as for the heterodimer of BRCA1 (breast cancer 1) and BARD1 (BRCA1-associated RING domain 1), Cullin–RING ligases, and APC [34]. The HECT E3s have a catalytic cysteine residue that can form a thioester bond directly with ubiquitin [34]. While the HECT domain represents the catalytic domain, the substrate specificity of HECT E3s is determined by their respective N-terminal extensions. The RING-between-RING (RBR) E3s define a third class of ubiquitin ligases distinct from the RING and HECT types. The RBR E3s are characterized by RING1 and RING2 domains and a central in-between-RINGs (IBR) zinc-binding domain. Substrate ubiquitination by RBR ligases is a multistep process. It starts with the recognition of the ubiquitinated E2 by RING1, followed by the transfer of ubiquitin onto the catalytic cysteine in RING2 to form the thioester intermediate, and finally, the transfer onto the substrate [35].

Most E3 ligases are able to generate different types of ubiquitin chains, depending on the E2 conjugating enzyme with which they preferentially interact. It was shown that the monoubiquitinating E2 enzymes have stronger affinity to RING E3 ligases. For example, UBE2A binds the RING E3 ligase RAD18, whereas UBE2K and UBE2T have high affinity to the RING E3 ligase BRCA1 [36,37]. Furthermore, a process termed coupled monoubiquitination is responsible for the monoubiquitination of ubiquitin-binding domain-containing proteins. The proposed model suggests that monoubiquitinated substrates cannot be further polyubiquitinated because the ubiquitin-binding domain interacts intramolecularly with the attached ubiquitin, thus disrupting the association of the substrate with the monoubiquitinated E3 enzyme [21]. Monoubiquitination can also be achieved by engaging a DUB that trims the assembled poly-ubiquitin chain to produce monoubiquitinated proteins.

Given that around 5% of the human genome encodes ubiquitin system components [38], it is not surprising that alterations of the ubiquitination machinery have been observed in multiple disease conditions [39]. Here, we will discuss how dysregulation of monoubiquitin signaling due to germline mutations of E3 ligases or DUBs could contribute to the development of genetic disorders (Table 1).

## 2. Genetic Diseases Associated with Dysregulated Monoubiquitination

### 2.1. X-linked Syndromic Mental Retardation

Germline alterations of the ubiquitin-conjugating enzyme E2 A (*UBE2A*) gene coding for the ubiquitin-conjugating enzyme 2A [40] are associated with X-linked syndromic mental retardation, a disease characterized by abnormal intellectual development and dysmorphic features, such as large head, wide mouth, almond-shaped eyes, and onychodystrophy. Patients with this syndrome carry intragenic point mutations or microdeletions of *UBE2A* or larger Xq24 deletions encompassing the *UBE2A* region [45,46]. The *Ube2a* knockout mice present defects in spatial learning tasks, but no other severe phenotypes [81]. This indicates that the mouse model only partially recapitulates the phenotype observed in patients suffering from *UBE2A*-linked mental retardation.

Molecular mechanisms linking *UBE2A* mutations to neurodevelopmental disorders are not fully understood. A study focused on the fly *UBE2A* homolog *Rad6A* implicates defective mobilization of the E3 ligase PARKIN as a cause of abnormal vesicle trafficking and dysregulated clearance of dysfunctional mitochondria in neurons. These abnormalities are suspected to contribute to the neurodevelopmental phenotype in patients with *UBE2A* deficiency syndrome [82]. On the other hand, one of the best-described functions of UBE2A is to promote monoubiquitination of proliferating cell nuclear antigen (PCNA) in a complex with the RING-Type E3 ubiquitin transferase RAD18. PCNA monoubiquitination can be switched to polyubiquitination in the presence of helicase-like transcription factor (HLTF). Two distinct branches of the DNA damage tolerance pathways are activated by either mono-, or polyubiquitinated PCNA to rescue a stalled replication fork and ensure continuous DNA synthesis. Monoubiquitinated PCNA favors low-fidelity translesion DNA synthesis, whereas PCNA polyubiquitination induces high-fidelity homology-dependent DNA repair [42]. Defects in DNA damage response could explain some of the developmental aspects of X-linked mental retardation [43,44]. *PCNA* mutations in patients also cause ataxia-telangiectasia-like disorder-2, a disease showing development delay [83]. Moreover, the disease-associated G23R mutation of UBE2A disrupts the binding site for RAD18 [84]. This suggests that the UBE2A/RAD18/PCNA axis might be at least partially responsible for the pathogenesis in mental retardation (Figure 1A).

In complex with the E3 ligase ring finger protein 20 (RNF20), UBE2A also promotes the monoubiquitination of histone H2B [41]. Monoubiquitinated H2B not only regulates global transcriptional elongation [85,86] but also plays an essential role in the regulation of inducible genes involved in cell differentiation [87,88,89] and inflammation [90]. However, a potential role of UBE2A-mediated monoubiquitination of H2B in mental retardation is still to be elucidated.

### 2.2. Parkinson’s Disease

PARKIN, or Parkinson Protein 2 (PARK2), is a RBR-type E3 ubiquitin ligase mutated in autosomal recessive juvenile parkinsonism [51], a form of familial Parkinson’s disease, defined by an onset before 40 years of age and characterized by slow movement and tremor (Table 1). *PARK2* is also mutated in other neurological diseases such as retropulsion, dystonia, hyperreflexia, and sensory axonal neuropathy [91] causing olfactory impairment [92]. In these different pathologies, loss of PARK2 function causes death of selective neuron populations, such as the dopaminergic neurons [93]. Deletion of *Parkin* in mice leads to motor and cognitive deficits [94] caused by catecholaminergic neuronal death and the subsequent loss of norepinephrine in some regions of the brain [95]. The *Parkin* knockout mice also show enhanced hepatocyte proliferation, macroscopic hepatic tumors in aged mice, higher sensitivity to myocardial infarction, and a strong inflammatory phenotype [96].

PARKIN maintains mitochondrial health through mitochondrial quality control and generation of mitochondrial-derived vesicles, followed by whole-organellar degradation, a process called mitophagy [97]. Mitophagy is vital for the removal of damaged mitochondria and toxic mitochondrial proteins, protecting neuronal cells from apoptosis [49]. Dysregulation of these processes plays a key role in Parkinson’s disease [50]. PARKIN was shown to mediate both polyubiquitination and monoubiquitination depending on the protein context [47]. This dual activity of PARKIN differentially affects function of its substrates such as voltage-dependent anion-selective channel 1 (VDAC1), which transports ions and small molecules at the mitochondrial outer membrane. Defect in VDAC1 polyubiquitination hinders PARKIN-mediated mitophagy, whereas dysregulation of VDAC1 monoubiquitination induces apoptosis. This suggests that the dual regulation of mitophagy and apoptosis by Parkin via VDAC1 poly- and monoubiquitination is critical in protecting cells from the pathogenesis of Parkinson’s disease [48] (Figure 1B). PARKIN also mediates the multi-monoubiquitination of heat shock protein 70 (HSP70) and heat shock cognate 70 (HSC70), leading to their association to insoluble substrates, consistent with a degradation-independent role for this type of ubiquitin modification [98]. These data strongly implicate PARKIN-mediated monoubiquitination in the development of Parkinson’s disease.

### 2.3. Fanconi Anemia

Fanconi anemia (FA) is a disorder caused by the genetic inactivation of crosslink repair. FA is characterized by abnormal development, bone marrow failure, hypogonadism, and marked cancer susceptibility. Autosomal recessive mutations in any one of 20 genes (*FANCA* to *FANCQ*) result in this genetic disorder, and collectively, the FANC gene products function in a FA−DNA crosslink repair pathway [99]. Several of the FANC genes form a large monoubiquitination complex (FA core complex). Ubiquitin conjugating enzyme E2 T (UBE2T, also called FANCT) and E3 ligase FA complementation group L (FANCL) are key enzymes in the FA core complex and are mutated in different subtypes of FA, causing FA-T and FA-L respectively [56,57] (Table 1).

Several animal models were generated to study the disease. *Ube2t* knockout in zebrafish leads to hypersensitivity to DNA damage and reversion of female-to-male sex [99], reflecting the hypogonadism phenotype occurring in FA patients. *Fancl* knockout in mice leads to decrease of fertility and defects in the proliferation of germ cells. Bone marrow cells isolated from the *Fancl* knockout mice were also hypersensitive to DNA crosslinking agent, mitomycin C [57].

The hallmark of FA is a high frequency of chromosomal aberrations caused by defects in DNA repair and hypersensitivity to DNA crosslinking agents in cells isolated from patients [100]. When DNA replication is stalled, the FA core complex is activated and monoubiquitinates the FA group D2 (FANCD2) and FA complementation group I (FANCI) heterodimer [52]. Monoubiquitinated FANCD2/FANCI heterodimer adopts a closed conformation, creating a channel that encloses double-stranded DNA [53]. The ubiquitin residue plays a key role in this process as it acts as a covalent molecular pin to trap the complex on DNA. Moreover, monoubiquitinated FANCD2 serves as a signal to recruit to the replication fork, DNA repair proteins that contain ubiquitin-binding motifs [101], such as FA complementation group P (FANCP, also called SLX4) and FA complementation group Q (FANCQ, also called ERCC), to remove the cross-linked DNA [54,55] (Figure 1C). DNA crosslinking repair is completed when ubiquitin-specific protease 1 (USP1) reverses the monoubiquitination of FANCD2 [102,103,104]. Interestingly, knockout of *Usp1* in mice also recapitulates FA phenotypes, including perinatal lethality, infertility, and crosslinker hypersensitivity. Finally, FANCD2 monoubiquitination could also be mediated by the ubiquitin ligase activity of the E3 ligase BRCA1 [58]. *BRCA1* is also mutated in a subset of patients suffering from Fanconi anemia D1 (FANCD1), identifying another key ubiquitin ligase in the pathogenesis of FA [59].

### 2.4. Charcot-Marie-Tooth Disease

Mutations of LRSAM1 (leucine-rich repeat and sterile alpha motif containing 1), an E3 ligase with RING ZINC finger domains and leucine-rich repeats, are associated with cases of early-onset Parkinson’s disease [105] and to Charcot-Marie-Tooth disease [62] (Table 1). Charcot-Marie-Tooth disease affects the peripheral nervous system and is characterized by progressive muscular atrophy. Knockdown of *Lrsam1* in zebrafish leads to disturbed neurodevelopment with a less organized neural structure and affected tail formation and movement. *Lrsam1* mutant mice present a normal neuromuscular structure and only mild neuropathy phenotype in aged mice, but higher sensitivity to neurotoxic agents that cause axonal degeneration [106].

Mechanistically, LRSAM1 promotes monoubiquitination of TSG101, a component of the endosomal sorting complex required for transport (ESCRT)-1. LRSAM1-mediated monoubiquitination of TSG101 enables recycling of TSG101-containing sorting complexes and cargo reloading [60]. Several proteins directly regulating post-translational processing and intracellular trafficking, such as tripartite motif-containing 2 (TRIM2) and RAS-related small GTPase RAB7, are also found to be mutated in Charcot-Marie-Tooth disease, confirming the link between vesicle trafficking and Charcot-Marie-Tooth disorder [61].

### 2.5. Cushing Disease

Mutations of the ubiquitin-specific peptidase 8 (USP8) are associated with pituitary adenoma tumors, also called Cushing disease [68] (Table 1). In Cushing disease, the adrenocorticotropic hormone (ACTH)-producing pituitary adenoma tumors secrete cortisol in the blood, resulting in obesity, diabetes, hypertension as well as additional cerebrovascular, cardiac, and reproductive disorders [107]. Disease-associated *USP8* mutations are located within or adjacent to the 14-3-3 binding motif. The diminished ability of mutant UPS8 to interact with 14-3-3 proteins enhances the proteolytic cleavage of USP8, leading to the generation of an activated catalytic fragment [69].

Enhanced activity of mutant USP8 in Cushing disease impairs the down-regulation of the epidermal growth factor receptor (EGFR) pathway due to increased EGFR deubiquitination. Consequently, sustained EGFR signaling in pituitary adenoma leads to enhanced promoter activity of the gene encoding proopiomelanocortin (POMC), the precursor of ACTH [63,64]. However, a subset of *USP8* mutations are not associated with higher EGFR expression, and mutations in *USP8* rarely occur in other tumor types, suggesting that USP8-dependent mechanisms other than EGFR up-regulation cannot be ruled out to be responsible for the pathogenesis of Cushing disease [66].

The role of USP8 in endosomal sorting complexes required for transport (ESCRT)-mediated trafficking of RTKs remains controversial [108]. USP8 contains an N-terminal microtubule interacting and transport domain, which has unveiled its potential to interact with charged multivesicular body proteins (CHMP), components of the ESCRT-III complex [67]. One of the proposed mechanisms is the regulation of multi-monoubiquitination of charged multivesicular body protein 1b (CHMP1B) [65]. CHMP1B, a part of the endosomal ESCRT-III complex involved in endosomal budding, is multi-monoubiquitinated in response to growth factor stimulation [65]. Deubiquitination of CHMP1B favors its assembly into a membrane-associated ESCRT-III polymer complexes and modulates the dynamics of endosomal sorting [65]. Further studies on the USP8 targets may shed new light on our understanding of USP8 contribution to membrane receptor trafficking and chemotherapy resistance in Cushing’s disease.

### 2.6. Noonan Syndrome

Noonan syndrome (NS) is a complex disease where patients present various clinical features including short stature, dysmorphia, and different cardiac defects [71,72]. Infants affected by the disease suffer from polyhydramnios, pleural effusions, or edema. Recent studies have linked NS to lung lymphangiectasis that is characterized by vessel dilatation and hemorrhages. NS is caused by germline mutations in genes that encode protein components of the RAS/mitogen activated protein kinase (MAPK) pathway. The vast majority of these mutations result in increased RAS/MAPK signaling [75].

Germline mutations of leucine zipper-like transcriptional regulator 1 (LZTR1) are associated with Noonan syndrome and familial schwannomatosis [71,109] (Table 1). Schwannomatosis is characterized by multiple ‘schwannomas’, benign tumors of the peripheral and spinal nerves [110]. *Lztr1* loss in mice results in perinatal lethality due to cardiovascular dysfunction [12,70]. *Lztr1* haploinsufficiency in mice partially recapitulates NS phenotypes, including heart malformation, craniofacial features, and bleeding abnormalities [70]. *LZTR1* loss in Schwann cells promotes their proliferation and dedifferentiation [12].

LZTR1 serves as a substrate adaptor for Cullin3 (CUL3) ubiquitin ligase complex that controls ubiquitination of the RAS GTPases [12,30,111]. LZTR1-mediated monoubiquitination of RAS at K170 attenuates RAS association with the membrane, inhibiting the RAS signaling pathway [12,30]. Disease-associated *LZTR1* mutations diminish either LZTR1-CUL3 complex formation or its interaction with RAS proteins, which impairs RAS ubiquitination. Dysregulation of RAS monoubiquitination results in the hyperactivation of the MAPK pathway and explains the role of LZTR1 in NS and schwannomatosis (Figure 1D). LZTR1 was also reported to regulate vesicular trafficking of VEGFR through CHMP1B ubiquitination. Loss of *Lztr1* reduces multi-monoubiquitination of CHMP1B, thus blocking the disassembly of the ESCRT complex and the trafficking of endosomal VEGFR. This ultimately leads to the activation of VEGF signaling and explains the bleeding abnormalities detected in NS patients [70].

The RING-type E3 ligase Casitas B-lineage lymphoma (CBL) is also found to be mutated in Noonan syndrome-like disorder, as well as early-onset juvenile myelomonocytic leukemia (JMML) (Table 1). JMML is a myeloproliferative disorder characterized by malignant transformation of the hematopoietic stem cell (HSC) causing oncogenic transformation of HSC-derived cells. Most common *CBL* mutations in patients affect the RING domains, such as the Y371 residue, as its phosphorylation is important for activation of the ligase function [76,77]. Deletion of *c-Cbl* in mice causes a decrease of B- and T-lymphocyte function causing lymphopenia [112], partially recapitulating JMML. *c-Cbl* knockout mice also present increased sensitivity to cardiac infarct [113], echoing the cardiovascular defects detected in NS patients.

CBL is a key regulator of internalization and turnover of different receptor tyrosine kinases, as it monoubiquitinates endosomal adaptors important for the recycling and/or degradation of the RTKs [114]. Loss of CBL results in endosomal accumulation of the RTK and uncontrolled activation of the downstream pathways [73]. One of the endosomal adaptors modified by CBL is SH3 domain-containing kinase-binding protein 1 (SH3KBP1, also called CIN85), which undergoes ubiquitination in response to EGF stimulation. After EGFR is internalized, monoubiquitinated SH3KBP1 recruits active EGFR to the multivesicular bodies (MVB) that targets EGFR for degradation. Loss of *CBL* leads to constitutive activation of EGF signaling [74]. The alterations in the trafficking and degradation of these signaling proteins lead to up-regulation of the MAPK pathway that could at least partially explain the mechanism by which CBL contributes to human disease (Figure 1E). Interestingly, deletion of the *SH3KBP1* gene in B-cells in mice results in impaired T-cell-independent antibody responses and abnormal B-cell receptor response, partially recapitulating JMML phenotypes [115].

### 2.7. Autoimmune Disorder

Mutations in the HECT-domain E3 ligase Itchy (ITCH) are found in a rare form of autoimmune disorder associated with facial dysmorphism, organomegaly, and developmental delay (Table 1). These patients also present severe autoimmune inflammatory cell infiltration of the lungs, liver, and gut [80]. Genetic analysis of the patients identified homozygosity for a frameshift mutation of *ITCH* caused by a 1-bp insertion, resulting in a truncated protein [80]. Mice lacking expression of the *Itch* gene develop a wide spectrum of immunologic phenotypes, such as lung and stomach inflammation, as well as hyperplasia of lymphoid cells and itching [116]. *Itch* deficiency in mice also renders mice resistant to TNFα-induced acute liver failure [117].

Different mechanisms have been proposed for the role of ITCH in human pathogenesis. ITCH was shown to regulate the stability and subcellular localization of multiple targets by mediating not only poly-ubiquitination, but also monoubiquitnation. ITCH-mediated monoubiquitination promotes degradation of the key regulators of T-cell anergy, phospholipase C-γ1 (PLC-γ1) and protein kinase C-theta (PKC-θ) [118]. ITCH can also suppress inflammation by controlling ubiquitination of tumor necrosis factor alpha-induced protein 3 (TNFAIP3, also called A20) [119]. Moreover, in T cells, ITCH mediates monoubiquitination of TGF-β inducible early gene-1 (TIEG1) [78]. Monoubiquitinated TIEG1 is translocated to the nucleus and triggers expression of Forkhead Box P3 (FOXP3), a master regulator of T cell function (Treg cells). *Tieg1* deficient mice are not able to suppress lung inflammation, indicating that deficiency in TIEG1 ubiquitination can explain ITCH-mediated pathogenesis in patients with autoimmune disorder.

Furthermore, ITCH regulates subcellular localization of survival motor neuron (SMN) by mediating its monoubiquitination [79]. Specifically, SMN is expressed mainly in the nucleus, where it accumulates in subnuclear structures such as the Cajal body. Dysregulation of SMN ubiquitination significantly impairs its co-localization with small nuclear ribonucleoprotein (snRNP) in Cajal body foci [79]. Importantly, SMN loss of function causes spinal muscular atrophy, a neuromuscular disease characterized by motor neurons degeneration, suggesting that the ITCH/SMN axis might be a major driver of muscular atrophy and dysmorphism (Figure 1E).

## 3. Discussion

Up to now, the role of non-degradative monoubiquitination in human disease has been relatively understudied, as focus was mostly falling on the role of polyubiquitination in proteasomal degradation. Recent emerging evidences have highlighted the key function of monoubiquitination in a wide range of cellular processes. The findings listed here only represent the most characterized enzymes controlling monoubiquitination. Nonetheless, it reflects the high prevalence of alterations of the monoubiquitin pathway in such a broad array of genetic disorders. This suggests that disruption of the monoubiquitin pathway may be a major force driving the pathogenic phenotypes of such diseases.

The abundance of the ubiquitin-related enzymes mutated in genetic disorders indicates that targeting the ubiquitin pathway might have a utility for a range of genetic diseases. However, at present, we lack detailed knowledge on how monoubiquitin signals are generated and how they are decoded by the cell. This is challenged by the diversity and complexity of the ubiquitin pathway. Moreover, monoubiquitinated proteins might not have been accurately identified, because polyubiquitinated conjugates are recognized more efficiently by anti-ubiquitin specific antibodies. This leads to underestimation of the pool of monoubiquitinated proteins present in the cell and challenges their characterization. The development of novel tools to purify monoubiquitinated proteins using high-affinity ubiquitin-binding domains and synthetic biology approaches to efficiently generate monoubiquitinated proteins overcoming these issues.

It is also worth noting that when looking at the few drugs that were developed to target the ubiquitin pathway, most are meant to inhibit its functioning. Several inhibitors targeting the ubiquitinating enzymes described in this review have been reported. Ubiquitin variants that block the E2-ubiquitin binding surface of the RING domain of CBL were shown to specifically inhibit the activity of phosphorylated CBL [120,121]. A high-throughput screening to identify ITCH inhibitors discovered that clomipramine, a common antidepressant drug, blocks ITCH autoubiquitination and affects the ability of ITCH to ubiquitinate its substrates [122]. Screening for the inhibitors of UBE2T/FANCL identified two compounds that sensitize cells to DNA crosslinking [123]. Pharmacological inhibition of USP8 was shown to effectively suppress ACTH synthesis in vitro without causing any significant cytotoxicity, indicating its potential for the management of ACTH hypersecretion in Cushing’s disease [124]. However, considering that the disease-associated alterations of the ubiquitin ligases and DUBs are mostly loss of function, inhibitors targeting these enzymes would not be beneficial. This indicates that there is a need to develop novel strategies for targeted therapies of genetic diseases [125]. Several screens identified compounds activating PARKIN ubiquitin ligase activity [126] and enhancing mitophagy [127], such as the compound described in patent WO2018023029. While no in vivo validation is available for this compound yet, this demonstrates the feasibility of identification of E3 ligase activators, opening novel therapeutic options for patients with genetic disorders.

This review collected multiple evidences that monoubiquitination is a highly relevant process in the pathogenesis of a wide range of genetic diseases; however, further research is necessary to identify specific entry points for therapeutic intervention of monoubiquitination-dependent signaling pathways.

## Figures and Tables

**Figure 1 ijms-21-03036-f001:**
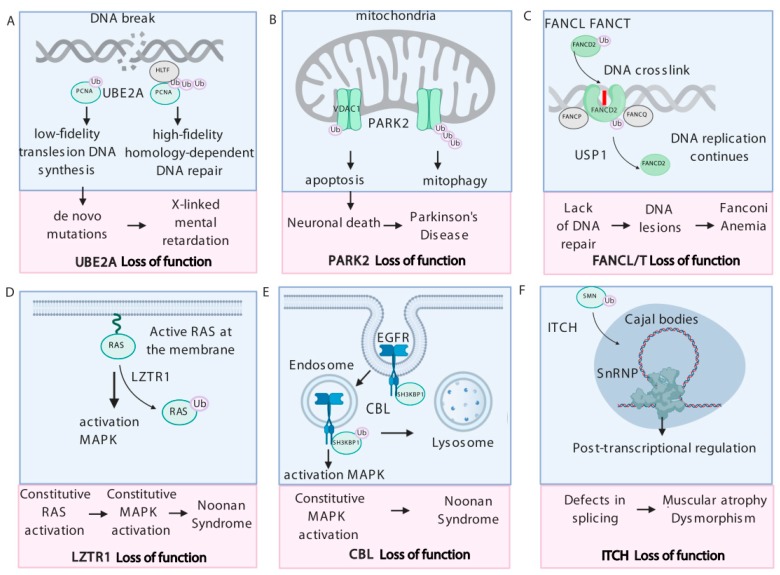
The role of monoubiquitination in human diseases. (A) Ubiquitin-conjugating enzyme E2 A (UBE2A) loss of function impairs proliferating cell nuclear antigen (PCNA)-mediated DNA repair that partially explains developmental aspects of X-linked mental retardation. (B) Parkinson Protein 2 (PARK2) regulates mitophagy and apoptosis by controlling poly- and monoubiquitination of voltage-dependent anion-selective channel 1 (VDAC1). Dysregulation of VDAC1 ubiquitination contributes to the development of Parkinson’s disease. (C) Mutations in Fanconi Anemia complementation group L/T (*FANCL/T*) lead to DNA repair deficiency. Monoubiquitinated FA group D2 (FANCD2)/FA complementation group I (FANCI) heterodimer binds DNA, whereas deubiquitination of FANCD2 allows to re-start DNA replication. (D) The Rat sarcoma (RAS) GTPases are monoubiquitinated by the Leucine Zipper Like Transcription Regulator 1 (LZTR1)- Cullin 3 (CUL3) complex, inhibiting RAS association with the membrane and activating of RAS signaling. Hyperactivation of the mitogen activated protein kinase (MAPK) caused by LZTR1 loss of function leads to the Noonan syndrome phenotypes. (E) Casitas B-lineage lymphoma (CBL)-mediated monoubiquitination of SH3 domain-containing kinase-binding protein 1 (SH3KBP1) recruits active epidermal growth factor receptor (EGFR) for degradation. *CBL* mutations lead to up-regulation of the MAPK pathway that partially explains its contribution to the development of Noonan syndrome. (F) Mutations in E3 ubiquitin-protein ligase Itchy (*ITCH*) impair the monoubiquitination of survival motor neuron (SMN) that dysregulates translocation to Cajal bodies and affects post-transcriptional regulation of gene expression, linking ITCH loss of function to the development of spinal muscular atrophy.

**Table 1 ijms-21-03036-t001:** Genetic diseases associated with genes regulating monoubiquitination. Short list of substrates modified by the indicated E2 conjugating enzymes, E3 ligases, and ubiquitin hydrolases (DUBs) are shown, together with the indication of the modulated cellular functions and the type of mutations detected in patients.

Disease	Gene	Type of Enzyme	Monoubiquitinated Substrate	Cellular Function	Disease-Associated Mutations
X-linked syndromic mental retardation	UBE2A	Ubiquitin-conjugating enzyme E2 A	PCNA [40]; Histone H2B [41]	DNA damage tolerance pathway [42,43,44]; epigenetic regulation [41]	Loss of function: missense mutations, microdeletions, larger deletions [45,46]
Autosomal recessive juvenile parkinsonism	Parkin or PARK2	RBR E3 ubiquitin ligase	VDAC1 [47,48]	Mitophagy, apoptosis [49,50]	Loss of function: missense mutations, deletions [51]
Fanconi Anemia	UBE2T	Ubiquitin-conjugating enzyme E2 T	FANCD2/FANCI [52,53]	Cross-linked DNA repair [54,55]	Loss of function: missense mutations [56]
	FANCL	PHD FINGER E3 ubiquitin ligase			Loss of function: missense, frameshift mutations [57]
	BRCA1	RING E3 ubiquitin ligase	FANCD2/FANCI [58]		Loss of function: missense frameshift mutations, deletions [59]
Charcot-Marie-Tooth disease	LRSAM1	RING E3 ubiquitin ligase	TSG101 [60]	Endosomal sorting [61]	Loss of function: missense, frameshift mutations [62]
Cushing disease	USP8	Ubiquitin specific peptidase 8	EGFR [63,64]; CHMP1B [65]	Endosomal sorting [66,67]	Gain of function: missense mutations [68,69]
Noonan Syndrome	LZTR1	BTB-Kelch ubiquitin ligase adaptor	RAS [12,30]; CHMP1B [70]	RAS localization and signaling [12,30]; VEGFR trafficking and signaling [70]	Loss of function: missense, frameshift mutations [71,72]
	CBL	RING E3 ubiquitin ligase	SH3KBP1 [73]	EGFR trafficking and signaling [74]	Loss of function: missense mutations [75,76,77]
Autoimmune disorder associated to facial dysmorphism	ITCH	HECT E3 ubiquitin ligase	TIEG1 [78]; SMN [79]	Nuclear translocation of FOXP3 [78], translocation of SMN to Cajal body [79]	Loss of function: frameshift mutations [80]

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
