# Peer review of "Cracking the Monoubiquitin Code of Genetic Diseases"

_ijms, 2020, doi:10.3390/ijms21093036_

Round 1
Reviewer 1 Report
The review by Sewduth et al. deals with the role of altered monoubiquitination of different substrates in disease. Monoubiquitination plays a critical role in numerous functions including trafficking, interaction between proteins, protein activity and stability, etc, of many relevant proteins associated to seminal physiological and pathological outcomes. Lack of monoubiquitination regulation in specific substrates leads to disease. This is a very well-written manuscript with a lot of information regarding many proteins that undergo monoubiquitination and multiple ubiquitination enzymes involved in the process. Although there is a very nice figure that summarizes some of the proteins and functions, it will be convenient also to include a Table that incorporates most of the proteins described together with the disease in which participate each one and the corresponding reference/s. This will further improve the quality of the review.
Author Response
- Although there is a very nice figure that summarizes some of the proteins and functions, it will be convenient also to include a Table that incorporates most of the proteins described together with the disease in which participate each one and the corresponding reference/s.
We appreciate the input from the reviewers. In the revised manuscript, we have summarized alterations of the monoubiquitination machinery in genetic disorders in Table 1.
Reviewer 2 Report
Sewduth et al. summarize biological involvement of monoubiquitination in human diseases. The authors started with a brief recapitulation of biology of ubiquitination with an emphasis on monoubiquitination. Followed are specific examples of involvement of monoubiquitination in selected diseases. The authors implicate that understanding monoubiquitination-related molecular events leading to various diseases could pave a way to development of therapeutics.
The manuscript is well written and seems to warrant publication in this journal in principle. Some minor points need to be resolved.
Line 2: genetic disease -> genetic diseases
Line 26: better to use "multi-monoubiquitination" in favor of "multi-ubiquitination" to distinguish events where monoubiquitination or polyubiquitination occur at multiple lysine/methionine residues.
Lines 119 and 126: references seem to be misplaced; in line 119 the reference [28] should be [27] and in line 126 the reference [29] should be [28]. Authors may want to sort subsequent reference numbering as needed.
Line 165: can the authors add a few sentences discussing HECT ubiquitin ligases given that RING and HECT comprise major categories of E3 ubiquitin ligases.
Lines 206 and 333: a space should be inserted between "the brain" and "[60]", and between "RTKs" and "[106]".
Author Response
Line 2: genetic disease -> genetic diseases
Line 26: better to use "multi-monoubiquitination" in favor of "multi-ubiquitination" to distinguish events where monoubiquitination or polyubiquitination occur at multiple lysine/methionine residues.
Lines 119 and 126: references seem to be misplaced; in line 119 the reference [28] should be [27] and in line 126 the reference [29] should be [28]. Authors may want to sort subsequent reference numbering as needed.
Lines 206 and 333: a space should be inserted between "the brain" and "[60]", and between "RTKs" and "[106]".
We apologize for these errors and have corrected them in revised manuscript.
Line 165: can the authors add a few sentences discussing HECT ubiquitin ligases given that RING and HECT comprise major categories of E3 ubiquitin ligases.
We have added more detailed information about different classes of the E3 ubiquitin ligases.